# Flexible Mobility On-Demand: An Environmental Scan

**Sohani Liyanage *, Hussein Dia *, Rusul Abduljabbar and Saeed Asadi Bagloee**

Department of Civil and Construction Engineering, Swinburne University of Technology, PO Box 218 Hawthorn, Australia; rabduljabbar@swin.edu.au (R.A.); sasadibagloee@swin.edu.au (S.A.B.)

* Correspondence: sliyanage@swin.edu.au (S.L.); hdia@swin.edu.au (H.D.);
  Tel.: +61-392-144-336 (S.L.); +61-392-145-280 (H.D.)

**Abstract:** On-demand shared mobility is increasingly being promoted as an influential strategy to address urban transport challenges in large and fast-growing cities. The appeal of this form of transport is largely attributed to its convenience, ease of use, and affordability made possible through digital platforms and innovations. The convergence of the shared economy with a number of established and emerging technologies—such as artificial intelligence (AI), Internet of Things (IoT), and Cloud and Fog computing—is helping to expedite their deployment as a new form of public transport. Recently, this has manifested itself in the form of Flexible Mobility on Demand (FMoD) solutions, aimed at meeting personal travel demands through flexible routing and scheduling. Increasingly, these shared mobility solutions are blurring the boundaries with existing forms of public transport, particularly bus operations. This paper presents an environmental scan and analysis of the technological, social, and economic impacts surrounding disruptive technology-driven shared mobility trends. Specifically, the paper includes an examination of current and anticipated external factors that are of direct relevance to collaborative and low carbon mobility. The paper also outlines how these trends are likely to influence the mobility industries now and into the future. The paper collates information from a wide body of literature and reports on findings from actual 'use cases' that exist today which have used these disruptive mobility solutions to deliver substantial benefits to travellers around the world. Finally, the paper provides stakeholders with insight into identifying and responding to the likely needs and impacts of FMoD and informs their policy and strategy positions on the implementation of smart mobility systems in their cities and jurisdictions.

**Keywords:** Flexible Mobility on Demand (FMoD); Mobility-as-a-Service (MaaS); shared mobility; Internet of Things (IoT); Cloud and Fog computing; sustainable public transport

---

## 1. Introduction

Providing access to high-quality urban transport services requires a variety of planning and operational innovations, as well as a better understanding of travel behaviour, operational processes, and the factors, which affect these issues.

Cities around the world are increasingly becoming a complex network of systems (also called super networks [1]) that are instrumented and interconnected in providing an opportunity for better management of vital services such as transport. An "Internet of Things (IoT)" is comprised of sensors and mobile devices all communicating with each other to enhance infrastructure capability and resilience and capturing volumes of data. Through data mining, artificial intelligence, and predictive analytics, smart city systems can provide travellers with more options to meet their mobility needs and allow city managers to monitor the performance of vital infrastructure and identify key areas where city services are lagging.

The convergence of physical and digital worlds is creating unprecedented opportunities to enhance the travel experience for millions of people every day through new mobility solutions driven by disruptive forces. Although some of these disruptive forces are still a few years away (e.g., autonomous vehicles), they have already started to shape a vision for a mobility transformation driven by converging forces including vehicle electrification, automated self-driving, mobile computing, on-demand shared mobility services, Big Data and predictive analytics [2]. The coming together of these powerful trends are shaping an urban mobility future inspired by a vision of low carbon living and zero road injuries. However, key to the success of these systems is a good understanding of the new business models, enabling technologies, and policies and regulations.

### 1.1. The Changing Landscape of Urban Mobility: Access Versus Ownership and the Collaborative Economy

The disruptive mobility trends have the potential to change fundamentally the relationship between the consumer and automobile. The rise of the collaborative or sharing economy, popularised by the companies such as Airbnb, Zipcar and Uber, has enjoyed remarkable rapid growth over the last few years and looks set to scale new heights over the next decade [3]. Access to mobility rather than to car ownership will enable customers to be more selective in choosing from the door-to-door mobility services offered by 'mobility operators' for intercity, suburban as well as 'last kilometre' travel solutions. Today, consumers use their cars as 'all-purpose' vehicles regardless of whether they are commuting alone to work or enjoying a leisurely drive with the family to the beach. There are already significant early signs that the importance of private car ownership is declining, and shared/collaborative mobility is increasing. With mobility offered as a service, consumers in the future will have the flexibility to choose the best solution for a specific trip purpose, on demand using their smartphones. According to a recent report [3], the shift to diverse mobility solutions is likely to result in 10% of new cars sold in 2030 to be a shared vehicle. This will reduce private-use vehicle sales but at the same time increase utilisation of shared vehicles. This will lead to a faster replacement rate of shared vehicles, which may offset some of the reduction in private-use vehicles. The report also estimates that more than 30% of kilometres driven in new cars sold could be from shared mobility.

### 1.2. The Need for Shared On-Demand Mobility Solutions

Today, with ever-rising urbanisation, most cities around the world are challenged to provide transport infrastructure and mobility services to meet the travel demands of increasingly connected populations [4–9]. The role of reliable public transport services becomes crucial as this leads to sustainable urban mobility where the movement of people, goods, and freight is safer and more efficient. However, conventional public transport services operate in fixed routes and schedules, which make them less attractive compared to personalised travel options such as private modes, which offer more convenience and flexibility. The low frequency during off-peak periods and in low-density areas also reduces the reliability of conventional public transport [10]. This leads to car-dependent communities, where there are higher numbers of low occupancy vehicles on the road. This leads to higher congestion, energy consumption, and increases in emissions and environmental pollution [11–13].

Flexible on-demand transport has increasingly become attractive as a sustainable and economically feasible alternative to conventional bus services [14–16]. Flexible Mobility on Demand (FMoD) systems, also known as demand-responsive transport (DRT), Dial-a-Ride Transit (DART), and flexible transport services (FTS), have been trialled with variable success across a number of cities around the world. In this paper, FMoD refers to both shared mobility solutions of passenger vehicles and also the emerging forms of on-demand public transport that use larger vehicles such as small buses. In the last decade or so, door-to-door transport services have evolved from DRT services, which operate as a service for elderly and disabled users who have difficulties in using conventional public transport [17–19], to the more sophisticated app-based services in operation today. Interestingly, these forms of shared transport are gaining more popularity. For example, a survey in the San Francisco Bay

Area showed that 60% of respondents were willing to accept these personalised services [20]. From a transport service operator perspective, fixed-route and fixed-schedule conventional bus services may appear to be cost-effective in urban areas, but they have been found to perform poorly in low-density areas [21] and outside peak hours. Hence, there has been some strong and renewed interest to explore how demand-responsive transport would be beneficial for both users and operators. This is reflected in recent research in this field, which has not only looked at trials but also at the potential benefits of FMoD as a form of future urban mobility.

FMoD can be thought of as a form of public transport which aims to provide a convenient service to users through advanced technologies [22]. It is a user-targeted innovative approach aimed at providing efficient, safe, and economical mobility options to passengers and users of the transport system [23]. Others think of it as the transformation of private automobile ownership models towards more flexible shared solutions [23]. To be successful, FMoD should integrate and connect transit networks and operations, in real-time data, and provide users with a variety of new travel options. This allows users to plan and book their trips, get access to real-time information and to process the payment through a single user interface [24].

In its early days, users had to call and place their request few days prior to the trip, and then operators manually scheduled the trip [17]. However, advancements in mobile technologies and communications have led to rapid advances in Flexible Mobility on Demand services [22]. Users are now able to request, place their booking online and track the vehicles in real-time via the smartphone application. The size of vehicles has also changed, making way for more compact vehicles suited for transporting smaller numbers of people. In the process, this had led to a blurring of the lines between traditional public transport (buses) and shared mobility services. The characteristics of this user-oriented public transport include flexible routing and scheduling of adequate fleets that need to be operated in the form of shared transport as per passengers' requests on pick-up and drop-off locations. Therefore, the main ethos of a successful FMoD scheme is shared services, reliability, affordability, accessibility, and on-demand In addition, these FMoD are flexible in terms of route choice, vehicle allocation, and payment type [25]. Specifically, the main principle behind the FMoD operation is that users share their rides with passengers who follow similar routes from origin to destination. In terms of accessibility, customers with special needs such as disabilities, mobility aids, or other additional assistance could request rides suiting their needs.

*1.3. Technological, Social, and Economic Impacts of Shared On-Demand Mobility Solutions*

This paper focuses on providing an overview of flexible on-demand transport with emphasis on the role of these services as a form of public transport. The paper also looks more broadly at the technological, social, and economic impacts of shared on-demand mobility solutions through a comprehensive environmental scan which included acquisition and use of information about the topic in addition to developing a good understanding of the trends and relationships related to the topic. The paper's finding will address the needs of different stakeholders involved in shared mobility including researchers, practitioners, and policy makers [26]. For the environmental scan in this paper, the main search terms used to gather the information included; Mobility-as-a-Service (MaaS), demand-responsive transportation (DRT), flexible transport services (FTS), Dial-a-Ride Transit (DART), Mobility on Demand (MoD), Autonomous Mobility on Demand (AMoD), Flexible Mobility on Demand (FMoD), customized bus service, full demand service, etc. The main research databases that were used to gather and collate the information included Google Scholar, Research Gate, Springer, Scopus, Science Direct, TRB, IEEE, etc. The paper also benefited from findings from national and international stakeholder workshops attended by a number of authors.

The remainder of this paper is organised as follows: Section 2 provides a detailed overview of Flexible Mobility on Demand (FMoD) systems. Section 2 is subdivided as follows: (i) the historical perspective: early beginnings, and (ii) the case studies—mapping the value beyond the hype. Section 3 summarises the impacts, benefits, and user acceptance of shared mobility. Section 4 underlying

the disruptive technologies in FMoD. The disruptive technologies in FMoD that are discussed in Section 4 are digitalisation and the internet of things, mobile computing, Internet of Things (IoT), cloud, fog and quantum computing, crowd-sourcing and data fusion, and data analysis techniques. The summarised data analysis techniques are linear regression, time series analysis, clustering and classification methods, machine learning, big data analytics, optimisation techniques, deep learning and reinforcement learning. Then Section 5 discusses the existing and emerging business models. In Section 6 the challenges and opportunities of FMoD are described. Section 7 describes policy insights and Section 8 provides a summary of this paper review and future research directions.

## 2. Flexible Mobility on Demand (FMoD)

This novel user-centric approach focuses on providing convenient mobility options. The goal is to enable a more efficient, safer, reliable, and smarter mobility to benefit individual travellers, transport system operators and managers within a multimodal ecosystem. The key advantage of FMoD is in providing users with enhanced travel options through integrated, efficient, user-focused transport system. Private and public transport providers also benefit through integrated and common mobility service platforms.

### 2.1. Historical Perspective: Early Beginnings

While recent advances and breakthroughs in technology have helped the push towards 'mobility as a service', the benefits have long been recognised [27]. Two examples of early experiments of collaborative mobility are presented next.

### 2.1.1. Purdue University—Mobility Enterprise (1983)

In January 1983, Purdue University began its Mobility Enterprise, which provided families who joined the scheme with a MAV (minimum attribute vehicle or 'small car') and access to a shared pool of larger vehicles. The aim of the project was to use the availability of shared cars to enable drivers to choose a fit-for-purpose vehicle for each trip, rather than having to use an all-purpose vehicle for every occasion. It was expected that this would increase the efficiency of the vehicle kilometres travelled and reduce fuel consumption. The experiment ran for two and a half years and incorporated an average of 12 households. To encourage people to join their experiment, the Mobility Enterprise had to be competitive when compared with private car ownership. The results from this initiative showed a significant reduction in fuel consumption, but no change in the driving habits of those involved. The researchers found that they were not able to influence the driving behaviour significantly of those involved. However, the project was able to demonstrate that fuel consumption could be reduced through the use of fit-for-purpose mobility and that a tangible monetary saving can be used to motivate change [27].

### 2.1.2. San Francisco—STAR Project (1983–1985)

The STAR (short-term auto rental) project was undertaken in the San Francisco area between 1983 and 1985. The project identified an apartment complex of 9000 residents that was well serviced by public transport, and established a car rental dealership within the precinct, which provided competitively priced short-term rental agreements. The aim of the project was to demonstrate that it was possible to live in an American City without owning a motor vehicle. The study found that since the public transport services were well utilised during the commute to and from work, many of the trips made by private vehicle were discretionary in nature. For these discretionary trips, the presence of a rental dealership would allow drivers the choice of a fit-for-purpose vehicle, providing fuel savings and increased vehicle utilisation. The project was a partial success, with those who made infrequent trips electing to forgo private vehicle ownership (and in some cases saving as much as $1000 a year). However, the project was limited in that it was only utilised for discretionary, non-work-related trips.

The project concluded that the scheme was well-suited to provide fit-for-purpose mobility in support of high-quality public transport [27].

## *2.2. Case Studies—Mapping the Value Beyond the Hype*

The ability to reduce congestion, cut travel times, improve trip efficiency, save petrol, and reduce pollution through the use of collaborative mobility has long been understood [27]. The addition of high funding levels and the potential to create an entirely new consumer market within the transport sector offers a tempting reward for anyone who can unlock the potential of collaborative mobility [3]. With the capacity for profit, combined with environmental and social benefits, a wide range of international initiatives has been launched, with varying degrees of success, in an attempt to harness its potential [28]. The following section will analyse some of the key collaborative mobility projects that have been undertaken around the world. It will focus on identifying success and failure factors and will try to identify lessons that may be learnt from each case.

In recent years, the convergence of technology and infrastructure has renewed interest in shared mobility. Some of the emerging trends in this space are covered next.

### 2.2.1. Car Sharing Services

As most private vehicles are only utilised less than 10% of the time, shared car services, such as Zip Car and Autolib, aim to increase vehicle utilisation and efficiency by maintaining a fleet of vehicles that can be accessed on either a subscription or hire basis [29]. Global car sharing provider Zip Car estimates that due to the increased utilisation of the vehicles, each of the shared vehicles can replace as many as 15 ordinary vehicles. Although it should be noted that most Zip Car users drive between 60% to 80% less than the average motorist and this allows each shared car to provide for a greater number of people that would otherwise be the case [30]. Surveys of the users of Autolib, the French car sharing operator, revealed that of those who did not own a private vehicle, 70% listed Autolib as the reason that they had been able to move beyond private car ownership [30]. In recent years, some shared car providers have been developing new additions to their services and forming new partnerships to further realise the potential of collaborative mobility [31]. In an attempt to increase the attractiveness of car sharing in inner-city environments, companies are attempting to provide services that facilitate short-term, impulse use of shared cars. BMW's Drive now program uses flexible hire services to accommodate short distance trips. Proving initially to be successful, the scheme has now been established in five German cities, as well as London and San Francisco [30]. Ford's city driving on-demand scheme follows a similar approach and utilises a pay-by-the-minute hire scheme with the option for one way trips [32]. In Germany, the car-sharing operator Flinkster reached an agreement with Ford to have access to Ford's existing network of car dealerships as depots for the car-share scheme. In doing so, Ford and Flinkster have created a nation-wide car-share network without the capital for purchasing land and storage facilities [32].

Two other schemes currently being trialled by Ford are for non-corporately controlled car sharing. The "Car Swap" register is an internal experiment where a registry has been compiled of Fords employees in the Dearborn area in the USA. The register includes people who are interested in car sharing their vehicles. If someone who is part of the scheme needs a vehicle with some particular feature, a search of the registry would provide a list of people who can contact to acquire the use of that vehicle. Ford's second user-managed scheme, "Share Car", is currently in the development stage, with a team in Bangalore partnering with Zoomcar to establish the legal framework that would allow a group of people to jointly share ownership of a private vehicle [32]. A final scheme from the Ford motor group is the Remote Positioning project being undertaken in Atlanta, USA. The project is aiming to perfect the use of drone vehicles that can be controlled from a centralised facility, as shared vehicles. The use of these vehicles would allow a shared car operator to both deliver a vehicle to the customer's door and retrieve used vehicles from any point without the cost and time involved in physically ferrying valet drivers backwards and forth between jobs [32].

### 2.2.2. E-Hailing

The largest, most heavily-funded and most well-known on-demand mobility providers are the e-hailing services such as Uber, Lyft, and Didi [30,33]. Using private drivers who supply their own vehicles (to minimise the capital cost associated with traditional fleet operations), these companies operate an on-demand taxi-like service in most major cities around the world [29,30]. In particular, Uber is rapidly destabilising the traditional taxi industry [30,33] with recent reports showing that together with competitor ridesharing companies, they have captured more than 50% of the taxi market share for trips. Uber also has a ridesharing variant of its service known as UberPool [34] which operates in a number of cities including London. Statistics released by Uber claimed that, after six months of UberPool being introduced to London, the system has saved 700,000 vehicle miles (1,120,000 km) or 52,000 litres of fuel [35]. This saving translates to savings of 124 tons of $CO_2$ [34]. Uber also claimed that in some cities, over half of the trips are now being made using UberPool [35].

### 2.2.3. Public Transport Innovations

While most cities are continuing to push ahead with the expansion of existing conventional public transport networks (Beijing alone constructed over 230 miles (380 km) of subway between 2008 and 2015), the world is increasingly seeking ways to diversify the current transport options [29]. Active transport (walking or cycling) has recently been gaining acknowledgement as a legitimate transport option, with bike share schemes flourishing around the world [29]. In Paris, these two trends have been combined, with the city including its bike share service under the umbrella of its public transport network [29]. Probably the most innovative approach to public transport in recent times is the on-demand buses such as Kutsuplus and Bridj described next.

*Kutsuplus—On-Demand Public Transport*

The city of Helsinki in Finland has been praised for its 'ambitious' plans and target to overcome the need for private car ownership by 2025. The Kutsuplus mobility experiment initiated is seen by many as the flagship project leading this attempt [36]. Hailed as the first true on-demand public transport service, the Helsinki Regional Transport Service ran a technology-driven minibus service which utilised advanced algorithms to assign vehicles based on passenger demand on a real-time basis [28]. The initiative entered active service in 2012 with a fleet of three dedicated minibuses. Kutsuplus allowed commuters to specify an origin and a destination point (within a defined service area), and the algorithm then identified a minibus travelling in that direction and instructed its driver to pick up the new passenger [37].

The Kutsuplus experiment was launched with two initial goals: (1) test the technological feasibility of using computer-based routing algorithms to overcome the difficulty of maintaining effective control over mobility, and (2) measure public support and willingness to pay for on-demand public transport [28]. The service received strong public support with positive feedback and continual growth in ridership figures [28]. A survey of user satisfaction recorded an overall satisfaction rating of 4.7 out of 5, 10% higher than the satisfaction rating received by Helsinki's conventional public transport networks [37]. An important finding from this experiment, which is relevant to other collaborative mobility operators, is that people were willing to accept longer travel times during a journey more readily than they would accept longer waiting times at the start of a journey. This finding helped the operators to revise the algorithm used for bus routing to minimise waiting times on pickup which resulted in a considerable increase in the number of users [37].

Although the cost per trip was reported to be lower than that seen in conventional bus services [37], the Kutsuplus bus service was deemed a financial strain on the public purse and was discontinued at the end of 2015 [38]. This was contrary to public expectations of seeing the service expanded into the future [28]. To increase ridership, the system needed to provide a more attractive service covering a larger service area, longer operating hours, and shorter waiting times at pick-up. While some improvements in service quality were achieved, large-scale increases in the services required the

purchase of extra vehicles [37]. Data collected during the trial showed that every time the service was expanded, the increase in passenger numbers improved the efficiency of the entire system because a higher density of users allowed the computer algorithm to plot a more optimised route with less meandering to pick up lone passengers [37]. To that end, it was reported that a 31% increase in the system's capacity would result in an average of a 60% increase in income [38].

*Other Examples of On-Demand Public Transport*

The near-success story of Helsinki's on-demand minibus service has drawn interest from around the world [38]. Kutsuplus demonstrated to both governments and individuals that on-demand collaborative mobility was challenging but feasible from a technology perspective [37]. Since Kutsuplus' launch, cities around the world have begun to adopt similar on-demand shared mobility solutions. Many of these schemes are directly modelled after Kutsuplus, like Via Transportation and Chariot systems [29]. Other initiatives, however, have used Kutsuplus as a base framework but have then tailored the system to overcome identified weaknesses and meet the needs of the operator [38]. An example is a Bridj service, which includes a number of cities and project in the U.S. including Kansas City (Missouri) where Bridj collaborated with the city to provide a partial on-demand service. The city-owned Bridj busses follow a rough route in accordance with an established timetable (like in a traditional bus service) but will respond to the customer request for pick up along that route [38].

In both London and New York, Ford has launched Dynamic Social Shuttle, its own on-demand minibus scheme with a focus on minimising response time. Ford aims to use this project to gain data on shared mobility patterns and better understand the social dynamics that are driving shared mobility [32].

The Split project in Washington DC uses the same software as the Kutsuplus mini busses to optimise vehicle routes and travel times. However, it has succeeded in overcoming the growth difficulties that crippled its predecessor by using private vehicles for its fleet. This Uber-style approach where drivers provide their own vehicles reduces the financial barrier to growth which prevented Kutsuplus from achieving the scale of operations that would have allowed it to function effectively [32].

In Bangalore, rather than using dynamic pricing to affect the cost of bus fares throughout the day, the bus operators issue a raffle ticket to every person who travels on a bus outside peak hours, with cash prizes for the lucky winners. The scheme doubled the pre-peak ridership and reduced peak hour travel times by 24% at little cost to the bus operator [30]. As a point of comparison, Singapore's attempt to influence pre-peak ridership on their rail lines achieved a 7% shift in passenger numbers by making all travel before 8 a.m. free [30].

## 3. Social Impacts: Benefits and User Acceptance of FMoD Systems

In past decades, sharing a vehicle with unknown passengers was not popular. Today, there is a significant positive psychological change toward shared mobility. This has partly been encouraged through sharing economy models in mobility, which has facilitated new transport solutions such as car sharing, ride sharing, bike sharing, and ride sourcing [39–41].

### 3.1. Car-Sharing

In recent times, car sharing has been promoted as a viable alternative to car ownership. This is a model where customers become a part of a car-sharing program upon paying an annual subscription through which they will gain access to a fleet of vehicles. Users are charged based on the time they have used the vehicle away from the server pool. Moreover, gasoline, maintenance, and insurance fees are included in an hourly payment fee [42]. The car-sharing applications around the world can be categorised into three types: round-trip car-sharing; one-way car-sharing; and personal vehicle sharing. Car sharing's success includes operation schemes through 26 countries in 1100 cities worldwide and nearly millions of users in the past two decades only [43]. The first formal car-sharing companies have started in 1980 in Switzerland and Germany. Nowadays, car sharing in Switzerland has around 60,000

users in 900 locations with 2000 vehicles while there are about 75 companies with 40,000 members approximately in Germany. Recently, Car2Go started in Germany has expanded to 18 cities around the world with over 350,000 customers. This growth also reached the Netherlands, Austria, Sweden, and France. In 2008, the US, Europe, and Australia have reached 150,000 users of car sharing [44]. Car sharing is an integral part of the big picture 'shared mobility' vision, which enables access to services without owning them [45]. The two car sharing schemes, Business-to-consumer (B2C) and peer-to-peer (P2P) operate through smartphone and internet platforms. B2C is referred to as free-floating car-sharing, where users are not required to return the car to the pick-up location (i.e., a one-way trip is allowed). While P2C is where existing car owners rent their vehicles to others for a short period of time [41,46].

Many studies were conducted to study the effects of sharing a car, owned vehicle usage and the distance travelled. In a study in Bremen and Belgium [47], the authors found car-sharing systems have replaced 7–10 private cars (in Bremen) and 4–6 cars (in Belgium). In another study [48], it was shown that 30% of users have sold a car or delayed purchasing a new one in San Francisco, USA. Similar to Chicago, where the I-Go Car Share system has also decreased the purchase of private cars [49]. In Switzerland [50], there were reductions of 33–50% in car kilometres travelled and an increase in public transport usage after joining the car-sharing schemes. In Germany, car sharing reduced the number of private vehicle users by approximately 50% while in Ireland, public transport and alternative travel modes (such as bicycles, trains, buses, and walking) increased due to car sharing systems.

The advantages of car sharing compared to car rental services and private car ownership was also discussed in [47]. In comparison to car rental services, car-sharing companies attract users by offering features for more suitable access to the car such as convenient location and flexibility. Compared to vehicle ownership, care sharing is comparatively easy to use and its main goal is to replace private vehicle ownership and reduce the number of privately-owned cars. It is considered a good alternative for owning a car, particularly for people who drive less.

Surveys have also shown key considerations why people are encouraged to use car sharing. For example, in Singapore, these reasons included cost savings, flexibility in managing family trips without investing in a second car, or financial inability to own and maintain a car [51]. The same study discussed the car-sharing user satisfaction, where it was found that 76% of users are satisfied with the car sharing in Toh Yi and 50% in Bishan. In addition, the main features that increased the level of user satisfaction are the availability of the car, possibility to choose the desired car from a number of available options, convenient booking and payment arrangements, and the good condition of the car [51]. It was also shown [52] that the main reasons for using car sharing schemes in the UK include cost-effectiveness, reducing emissions foot-print, and users believing that they are making a difference by sharing.

### 3.2. Bike-Sharing

Bike sharing is the shared use of a bicycle fleet, which is delivered by a commercial provider. The principle behind this concept is that individuals use bicycles based on an 'as-needed' basis without having the responsibilities of ownership and recurrent maintenance and operations costs [53]. Bike sharing has gained popularity in recent years with an increasing appetite for sustainable modes of transport. It was also facilitated by increasing ease of use through a mobile app and their value in the first and last mile connection to other modes of transit [54]. Key features of bike sharing schemes include [55]:

1.　Bikes can be rented from a particular location and could be returned either to the same place or to a different location.
2.　Easy to access with diverse business models underpinned by the use of technology such as mobile phones and smart cards.
3.　Provides first and last kilometre solutions that can be integrated with public transport systems

The largest bike-sharing scheme was launched in Paris with more than 20,000 bicycles in 2007. Today, China has the largest bike sharing scheme with over 70,000 bikes in Wuhan and 65,000 in Hangzhou [56]. New York City had a bike-sharing program with 10,000 bikes in 2013. Bike sharing is more convenient than the private bicycle as shown in a study conducted by [57]. The study shows that 60–70% of individuals in China believed that bike sharing is a more convenient option than using private bicycles. Similar results were found in Washington, DC, Minneapolis, and Melbourne [58,59]. Another study into bike sharing in North America was carried out by [59]. The study assessed bike sharing from both the operator and user point of view. The focus was on the impacts of bike sharing on other transport modes, user opinions and preferences, and the influence of travel distances on using bike sharing. The results showed that bike sharing has reduced walking, taxi, and car usage.

Currently, bike-sharing systems are used in more than 15 countries in over 75 cities around the world using more than 70,000 bikes. Major countries with a bike sharing system are Australia, Canada, China, France, UK, and the USA. Out of these, Europe's largest and most successful system, Velib, is in Paris [53,55]. Also, in Wuhan and Hangzhou, China operates the world's largest public bike-share schemes with over 65,000 bikes [60]. In addition, an online "capital bike-share customer use and satisfaction survey" was carried out in Washington DC in 2012. The results showed that 7% of 5464 of respondents were willing to shift from car usage to bike sharing [61]. Another survey in London found that 60% of the respondents were willing to shift to bike sharing [62]. There are many recent studies by Elliot Fishman on bike-sharing systems, Table 1 summarises the impacts of those relevant studies [63–67]. Other issues impacting their uptake include topography and climate in certain cities which do not suit bike sharing [68,69].

**Table 1.** Recent studies and their impacts on bike-sharing systems [63–67].

| Study | Data | Methods | Results | Recommendations |
|---|---|---|---|---|
| Risk Associated with Bike Sharing | Data collected from a Hospital to assess bike share and non-bike share cities in USA | Statistical approach | Lower risk of cycling injuries when bike sharing introduced | - A reliable methodology/tool Is required to collect data and reports related to the safety of Bike sharing schemes<br>- Bike sharing should be introduced with safety measures ensured by decision makers such as traffic calming |
| | Data for severe injuries collected from North American and European cities | Statistical Packages for the Social Sciences (SPSS), a Chi-square test and Incidence Rate Ratio (IRR) based on Poisson-regression with generalised linear models in SPSS | Low risk of fatal and severe injuries of a cyclist after a comparison between general and private biking schemes. | |
| Safety of Electric Biking and Classic Biking | Data on crashes collected from victims treated at emergency departments (ED) and information on cyclists for electric bike (EB) and classic bike (CB) users In Netherlands | Binary logistic regression | - EB users are involved in single biking incident while CB users are more likely involved in collision with other road users.<br>- Frequent bike users (4–7 days a week) aged 50–69 and CB users are more often treated at ED than other users. | More research needs to be conducted to study the impact of electric biking on road safety. |
| Factors Affecting Usage of Bike Sharing Schemes | Online survey for Bike sharing members and non-members in Melbourne and Brisbane | Logistic regression model | - Accessibility of bikes.<br>- Provide more stations near the workplace especially for employees under 35 years old.<br>- Increase bike lanes to increase safety will increase membership.<br>- Users with membership have a higher income than non-membership users since stations are located in the city area, not the suburb. | More studies are required on how to attract more users (in terms of all income levels) to become members in bike sharing programs Further research on the impact of compulsory helmet legislation on the usage of bike sharing. |

**Table 1.** *Cont.*

| Study | Data | Methods | Results | Recommendations |
|---|---|---|---|---|
| **Impact of Public Transportation System on Bike Sharing Mobility Pattern** | Bike sharing trips in London | Time series spatial and temporal analysis of bicycle trip counts and durations | - 85% increase in bicycle trip counts and 88% increase in trip duration because of public transport systems distributions<br>- Improve connectivity of network and enhance interactions between biking stations.<br>- Bike sharing can reduce reliance on public transport during disrupting events. | Quantitative and qualitative data needs to be collected on public transport and bike sharing users and more research is required to analyse their behaviour. Interactions between public transportation systems and bike sharing schemes are important and should continue to include integration of ticket systems and locations of bikes' stations. Hence, it will help to plan mobility options ahead during disruptive events. |
|  |  | Complex network analysis: in which each bike station is a node and a link is formed between two stations |  |  |

## 3.3. Mobility-as-a-Service (MaaS)

The drive towards collaborative mobility has received strong support, with the transport sector receiving the highest levels of funding for any sector within the collaborative economy [70]. This is manifested by one of the most promising trends within the disruptive mobility space, known as 'Mobility-as-a-Service' or (MaaS). The key concept behind MaaS is to place the road users at the core of transport services, offering them mobility solutions based on their individual needs. This can be achieved by providing a single platform for combining all mobility options and presenting them to the customer in a simple and integrated manner. This means that easy access to the most appropriate transport service will be included in a bundle of flexible travel options for customers. MaaS has the potential to fundamentally change the behaviour of people and reduce reliance on car ownership by providing easy on-demand access to the mobility services they need (Table 2). The trend is therefore gradually shifting from the provision of buses, trams and trains to a focus on what people require, and how a more considered and integrated approach could produce better outcomes [30].

**Table 2.** New urban mobility services (adapted from [29]).

| | **Traditional Mobility Solutions** | **New Mobility Services** | |
|---|---|---|---|
| **Individual-Based Mobility** | Private car ownership | Car sharing: peer to peer | A peer-to-peer platform where individuals can rent out their private vehicles when not in use (e.g., Turo) |
| | Taxi | E-hailing | Process of ordering a car or taxi via the on-demand app. App matches the rider with driver and handles payment (e.g., Uber, Lyft, Didi) |
| | Rental cars | Car sharing: fleet operator | On-demand short-term car rentals with the vehicle owned and managed by a fleet operator (e.g., GoGet, Car2Go, ZipCar, Getaround) |
| **Group-Based Mobility** | Car pooling | Shared e-hailing | Allows riders going in the same direction to share the car, thereby splitting the fare and lowering the cost (e.g., UberPool, LyftLine) |
| | Public transport | On-demand private shuttles | App and technology-enabled shuttle service. Cheaper than a taxi but more convenient than public transit (e.g., Bridj) |

The key advantages to these new services include reliability, predictability, convenience, and ease of accessibility. Most services also offer easy and secure payment options using cashless mobile

transactions. For a given trip, consumers can select from different types of services based on trip distance, waiting and travel times and levels of service [71].

Another key feature of MaaS is that it combines the use of public and private transport solutions to provide near door-to-door service [72,73]. It uses different modes of transport to deliver a tailored mobility package including complementary services such as trip planning, reservations, and payments through a simple mobile application [74]. A single interface is provided through which customers create and manage their total journey, with payment to all service providers coming from a single account and a single payment. Therefore, the main concept behind MaaS is to provide mobility solutions to travellers [75,76] based on their requirements. The scenarios for future urban mobility and the way to get there are schematically shown in Figure 1.

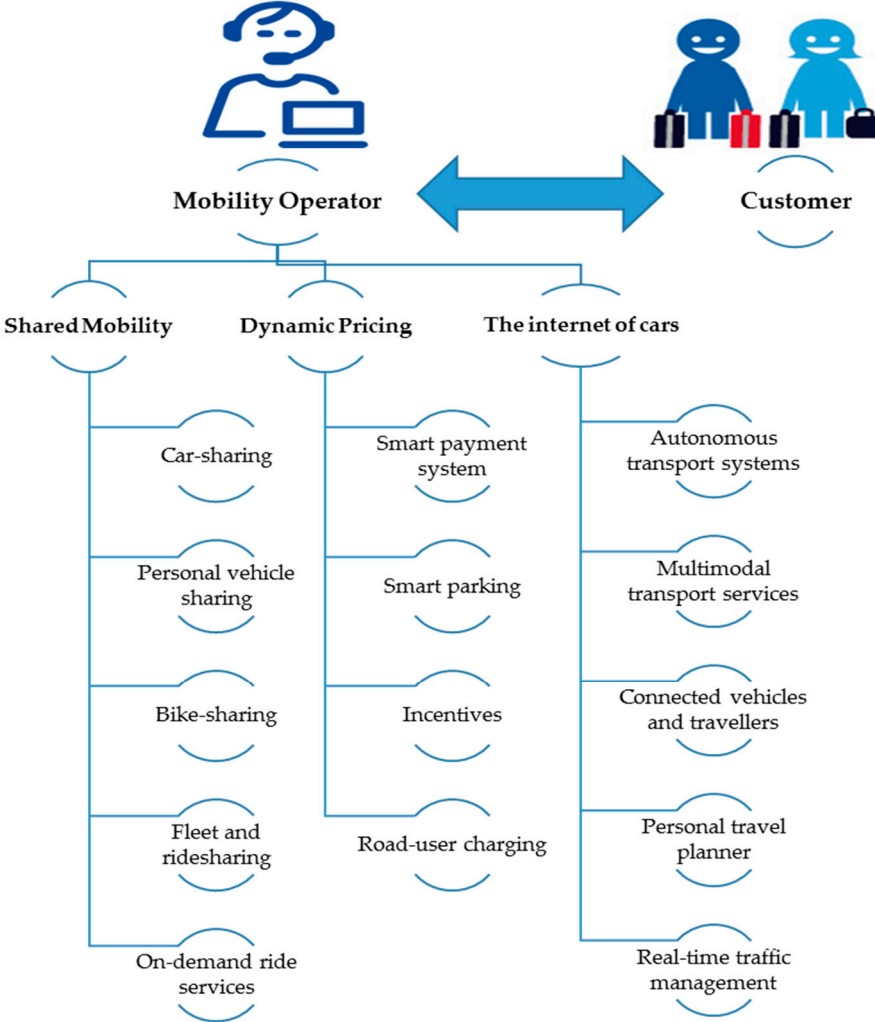

**Figure 1.** Digital-age urban mobility options (adapted from [75]).

The MaaS model offers opportunities to enhance public transport to cater to customer expectations through a connected and integrated transport system [77]. A connected transport system integrates transport modes with the overarching goal of providing unified door-to-door service to the passengers. In the past few years, MaaS initiatives grew rapidly especially in Europe, and some of these existing travel services or initiatives offered to customers through monthly subscription schemes include MOBIB in Brussels, Belgium [78], HANNOVER Mobil [79], EMMA in Montpellier, and SMILE in Vienna [80]. When implementing the MaaS concept, it is important to identify the factors that are

important to set up a reliable, working and successful scheme. Some of the core characteristics of MaaS implementation [19,81–86] are listed in Table 3.

**Table 3.** Characteristics of MaaS implementation [19,81–86].

| | Characteristics | Description |
|---|---|---|
| 1 | Integration of transport modes | This allows bringing together multi-modal transport systems to allow users to choose and facilitate their intermodal trips. Such as public transport, taxis, car/bike/ridesharing, and on-demand bus services. |
| 2 | Tariff option | MaaS would allow users to choose from two options, "Mobility package," and "Pay-as-you-go" to pay the tariff. These would be modelled based on kilometres travelled, minutes spent, and points collected on travels. |
| 3 | One platform | MaaS relies on a digital platform that allows users to do trip planning, booking, payments, ticketing and acquire other necessary information such as weather details. |
| 4 | Multiple actors | The system is built upon the interaction between different parties through a digital platform. They are end-users of mobility services, service suppliers, platform owners, and external stakeholders like local authorities, telecommunication, data management companies, and payment clearing agencies. |
| 5 | Use of technologies | MaaS is enabled through the integration of technologies such as Wi-Fi, 3G, 4G, LTE, and GPS, e-ticketing, e-payment, Internet-of-Things (IoT) and database management systems through mobile devices or computers. |
| 6 | Demand orientation | Best possible transport solution is offered based on the perspective of the customer via a multimodal trip-planning feature. |
| 7 | Registration requirement | Customers need to have registered with the system to access the available services. This either could be a personal account or for an entire household. The subscription also allows users to personalise their travel demand profiles and preferences. |
| 8 | Personalisation | Users requirements are made more efficient based on the uniqueness of different customers. The system would provide tailor-made and specific solutions to the users based on their profile, preferences, and history records. |
| 9 | Customisation | This enables customers to change/modify the offered services according to their needs. This is a key characteristic and attractive feature of MaaS. |

Furthermore, MaaS can include other attributes such as gamification where customers are rewarded for taking environmentally friendly trips; access to parking, park and ride services; and subscription plans based on customers' monthly travel budgets [87–89]. Parking pricing can come very handy to encourage Maas in a more effective and efficient way [90].

MaaS solutions benefit users in numerous other ways, especially providing customers with new flexible transport modes, such as point-to-point (taxi + public transport) or point-via-point-to-point (ride sharing + public transport). Increasingly, these flexible transport solutions could pose a threat to conventional public transport, particularly suburban buses.

To enable personalised services with seamless trip chains, it is important to have integrated transport modes. The digital platform should also provide options to pre-book and pay for a trip package through a mobile application.

Another important aspect to highlight is that MaaS solutions provide customers with transparent costing of trips. Many consumers do not have a complete understanding of the actual costs of travel

in private vehicles. However, with above-mentioned mobility services including car sharing, users can compare the traveling cost and determine which mode is more economical to use. For example, a case study in Finland conducted in 2011 showed that car-related travel accounted for 80% of the total cost of transport per household [91]. Flexible choices such as car sharing and on-demand ride-sharing services would be an effective replacement for private cars. The study also found that for shorter travel distances, walking or cycling would be more effective.

*3.4. Environmental Impacts*

Car dependency is a major contributor to environmental emissions in cities [11–13]. Melbourne is one example where 75% of the trips are made by car [92]. Passenger vehicles account for consuming 18 million barrels of oil each day which result in 2.7 billion tons of carbon dioxide annually, worldwide [24]. Low occupancy cars also contribute to traffic congestion resulting in many detrimental effects such as increased fuel consumption, higher risks of accidents, more vehicular emissions, and increased transport costs [93]. In Melbourne, urban traffic congestion accounted for approximately $3 Billion in 2005 and is predicted to double by 2020 [94].

Therefore, public transport services are essential to address the above-mentioned environmental and cost issues. Shifting passengers from low-occupancy vehicles such as single-passenger private vehicles to shared transport is one of the key strategies pursued by city decision makers around the globe. Car-pooling, for example, is a powerful strategy for delivering passenger trips that have similar origins and destinations. This is beneficial for both users and the wider community and can essentially be considered as a form of public transport. At the individual level, users will have a reduction in the cost of petrol and maintenance of their cars while the city and community will have reduced traffic congestion and pollution [95]. In a study conducted to identify the economic effects of a free-floating car sharing system in Germany [96], the authors analysed data from the car2go car-sharing scheme. This scheme allows users to take and leave vehicles at any point within the city and does not require prior booking and users are charged a fixed 0.19 €/min (0.25$/min) for their journeys. The survey indicated a reduction in $CO_2$ per average car2go user and a contribution to reducing private vehicle usage in the cities. In another study for Singapore [51], the authors showed that car sharing is more affordable for customers who are charged based on time and distance. The car-sharing scheme was also found to have contributed to reducing traffic volumes.

Furthermore, [96] noticed a major change in emissions as a result of the Car2go system in Germany. The study looked at the lifecycle of vehicles starting with manufacturing, raw material consumption and energy usage. It also looked at the operation of the system including emissions and fuel consumption. Emission results included a reduction of total emissions CO, $CO_2$, NOx, and SOx with a further reduction in traffic-related noise. In addition, [97] further stated that there could be a 5–8% reduction in fuel consumption with the implementation of ride sharing.

Autonomous vehicles are also being promoted as a form of future public transport. A number of studies have examined the advantages of the use of shared autonomous vehicles (SAVs) including environmental benefits and cost efficiency. One study [98] found that total kilometres of personal car travel would be reduced leading to a reduction in private car purchases. In another study [99], it was estimated that SAVs are more cost-efficient as they are expected to reduce trip costs by one to eight dollars per mile (on average) due to automation and removal of human labour. Another study [100] which looked at the impacts of SAVs on environmental sustainability, the authors found that each SAV has the potential to replace approximately ten privately owned vehicles. They also predicted a reduction in energy consumption, greenhouse gas (GHG) emissions, and air pollutants emissions. Thus, evidence from multiple studies suggests that car sharing (current and emerging forms) is environmentally feasible and economical.

Therefore, the implementation of a flexible on-demand public transport system, which encourages people to minimise the use of low occupancy vehicles, would be environmentally feasible and cost-effective in the long run.

## 4. Technology Trends and Developments—Underlying Disruptive Technologies in FMoD

The success of FMoD systems is due to a number of converging factors including the fast pace of development of disruptive technologies, which are facilitating the wide adoption of these mobility solutions. Some of these key technologies are described next.

The shift towards embracing collaborative mobility is driven by the recognition of the potential opportunities, and risks generated as a result of the interaction between existing and emerging disruptive technologies [3]. The use of collaborative mobility has been seen as a way of harnessing new developments, like big data analytics and the Internet of Things, to mitigate against the effects of potentially negative trends, like increased congestion and pollution resulting from continued urbanisation [27,101].

With population growth expected to add an additional billion people to the world's inhabitants within the next decade, and with 60% of these people expected to live in cities, transport operators are faced with a situation where traditional methods will no longer be sufficient to restrain congestion [29,102]. Under these conditions, increasingly large amounts of effort are being diverted into alternate means of providing mobility [33]. In addition to the efforts by transport authorities, who are already struggling to keep pace with existing congestion levels, the predicted migration of the earth's population towards its cities will cause congestion to spread to regions that have never before experienced traffic issues [29,38]. The following section reviews few of the opportunities and trends and identifies ways that they have the potential to influence the drive toward collaborative mobility.

### 4.1. Digitalisation and the Internet of Things

Since the conception of collaborative mobility, attempts to organise large-scale programs have been hampered by the difficulty of establishing an effective central administration [27]. The underlying premise behind any collaborative mobility enterprise is to increase transportation efficiency by increasing vehicle utilisation. This cannot be easily achieved without strong centralised control of the vehicle fleet [30]. Early examples of failed attempts included the Procotip (Montpelier, France, 1971) and Witkar (Amsterdam, 1973) shared car schemes. Both of these schemes made compact small vehicles available for use in and around the congested inner-city areas using a payment token system. Both of these schemes failed because of inefficiencies in vehicle distribution caused by a lack of control over the vehicle fleet, resulting in poor asset utilisation and excessive downtime between each trip [27].

At the same time, the advent of the 'Internet of Things'—whereby people and physical objects are constantly interconnected to the Internet—has also exacerbated effective control of such undertaking [30]. Operators are now provided with unprecedented levels of data, including real time analysis on everything from vehicle positions to weather patterns. With this data available, the challenge facing transport decision makers is no longer about gathering information, but how to use the information available for decision making in a meaningful and timely manner [103]. In Finland, the Kutsuplus bus network received constant updates on traffic conditions, allowing drivers to provide updated estimates of arrival times to their passengers, often accurate to within 30 s [37]. Around the world, software developers have developed tools to combine information on every transport option available in a given city. Users simply enter their origin and destination, and the software would provide them with a route optimised for cost, travel time, comfort, or environmental efficiency [36].

### 4.2. Mobile Computing

The 2018 global digital report reports the number of mobile phone users to have reached around 5.125 billion users, and 4.021 billion internet users, worldwide. People tend to use internet and mobile phones with GPS systems to get help with the navigation to their desired destination and to get information about real-time information about traffic and updates about the shortest or fastest routes. Mobile computing together with wireless communication and remote sensing are considered the key driving forces behind today's Intelligent Transport Systems (ITS) which are creating a huge leap

forward in future mobility. Vehicles embedded with sophisticated computer systems and on-board sensors have also allowed for collecting information about the environment, the location of the vehicle, and to exchange information with other nearby vehicles in real-time [104]. This has enabled transport systems operators to undertake a real-time analysis of travel patterns and establish adaptive management to operate transport systems more efficiently.

### 4.3. Internet of Things (IoT)

The use of IoT facilitates reliable IT-based infrastructure to provide an interconnection through of computing devices, which enables to exchange data in a secure manner. This has been possible due to the use of the internet of computing devices that are embedded into everyday objects that are allowed to be tracked, coordinated, and controlled across the internet or across a data network along with the use of sensors and actuators [105]. Currently, the use of IoT in transport is growing, as there are many potential areas of applications, such as developing a cloud platform to access a vehicular data, which enables the exchange of transport-related information [22]. Especially, information like vehicle location tracking and monitoring, road conditions and maintenance, and incident information, which are useful for traffic control and management. However, there are challenges to overcome with the developments of transport services through IoT, and the challenges are mainly in terms of security, privacy, scalability, reliability, lack of global standards, and service quality [22].

### 4.4. Cloud Computing and Fog Computing

Cloud computing can be categorised into public and private clouds. Public clouds include services offered by commercial providers such as Google App Engine and Amazon Elastic Compute Cloud. Private clouds are usually reserved services that are provided to specific end-users [106].

There are three main services enabled due to cloud computing including:

1. Software-as-a-Service (SaaS): The end-users are allowed to execute their applications remotely.
2. Infrastructure-as-a-Service (IaaS): The end-users are able to access the physical computing resources (computers, virtual machines).
3. Platform-as-a-Service (PaaS): Users are provided with tools and libraries by the cloud, to develop their own software.

In transport, advances in cloud computing and IoT are providing opportunities for better traffic management and customer-oriented services. As an example, to improve vehicle-to-vehicle communication and road safety, a novel vehicle cloud architecture named ITS-Cloud has been proposed [106]. Moreover, to optimise traffic control, cloud computing has been used to develop a cloud-based urban traffic control system [107]. However, cloud computing faces a number of challenges in the transport context, including cybersecurity and also establishing unified architectural frameworks with convenient functionality [106].

Cisco put forward a novel paradigm, called fog computing, which extends cloud-computing services to the edge of the network, and enables the communication, computation, and storage closer to edge devices and end-users. The benefits of fog computing include enhancing low-latency network connections between devices and analytics endpoints, reducing network bandwidth, fast data processing, security, and privacy [108]. Advances of fog computing mainly enable connected cars, smart cities and real-time analytics [109]. Reference [110] discuss the challenges that the existing traffic light control systems are facing, such as avoiding heavy roadside sensors, resisting malicious vehicles and avoiding single-point failure and how they have used fog computing to propose secure intelligent traffic light control schemes to overcome above-mentioned challenges. However, fog computing is challenged by security issues such as man-in-the-middle attack (MITM), i.e., an attacker actively eavesdropping the direct communication between two parties [111].

*4.5. Crowd-Sourcing and Data Fusion*

One of the key challenges facing mobility providers and data analysts is how to combine and fuse data streams from separate, often incompatible sources and combine them into a meaningful, user-friendly and cohesive set of information [112]. The computer algorithm used to direct the Kutsuplus busses was one of the first to combine user requests with spatial positioning and real-time traffic reports to control route planning and predict accurate estimates of arrival times [37]. The success of this computer algorithm has paved the way for a new era of informed journey planning [38]. By combining multiple information sources into one interface, tools like London's "CityMapper" application (now expanded to cover 10 EU cities) can increase the efficiency of people's mobility by allowing them to view all of their options in real time, and optimise their journeys [30,112]. Other initiatives such as Moovit and Moovel are also being used around the world with each system offering some unique features [30,113,114]. For example, the Moovit App augments its data collection by utilising customer feedback to increase the accuracy of its route recommendations [30]. The Waze App, while not a journey planning app, advises drivers of traffic congestion and disruption in real time, guiding their vehicles around hotspots to reduce travel time and save fuel [29].

*4.6. Data Analysis Techniques*

This section of the paper presents some of the tools that have been applied in FMoD applications and case studies. The discussion in the following sections will highlight the challenges and limitations of these approaches and outline directions for future research.

### 4.6.1. Linear Regression

Linear regression is a statistical analysis technique used to model the relationship between dependent and independent variables [115]. Regression techniques are well-known and have been used successfully to uncover information from dense transport data [116]. This includes recent studies where this technique has been used for data analysis as related to FMoD. For example, the effects of $CO_2$ emission changes due to the adaption of car sharing were investigated in [117]. First, mixed logit models were used to determine the preferences of using car sharing, and then binary logit models were used to determine if individuals are willing to shift from using their private cars to using car-sharing. Then, a linear regression model was used to analyse the behaviour of individuals and their social, economic, and environmental motives behind switching from using a private mode of transport to car sharing. Similarly, the factors that encourage private vehicle owners to shift to shared mobility were analysed using multilevel regression models [118]. Regression models have also been used in related applications to predict the travel times for different routes for public transport. Simulated data were used to predict average travel time for a bus route and overall bus travel times [119,120]. The independent variables considered for regression analysis included the length of the route, speed, frequency, flow intensity per Km, and the number of passengers. The results showed that it was valid to use regression models, but the models needed to be validated using field data.

Linear regression models were also used to predict the flow and demand of bike-sharing in Lyon [121]. In Greece, they were used to predict travel demand based on multiple linear regression analysis [122]. In another study, Harbin, the capital city of Heilongjiang province was selected to test the factors affecting people's choice of using customised bus service by using logistic regression (predictive analysis) [123]. Furthermore, another study used linear regression models to determine the number of passengers by usage and operational aspect in Great Britain [124]. In general, regression analysis has been used extensively in the past as a tool for data analysis, although in recent times there has been a noticeable shift towards applications of AI for data analysis which is being facilitated by fast computation techniques that can be used to identify patterns among variables [125].

### 4.6.2. Time Series Analysis

Data sets, which include time-ordered sequences of observations, are defined as a time series. Generally, time series observations are correlated and most standard statistical methods based on random samples are not applicable. Instead, time series analysis is used. These include two approaches [126]: time domain and frequency domain approaches. In the time domain approach, an autocorrelation function (ACF) and partial correlation function (PACF) are used as time functions to describe the characteristics of a time series with an evolution represented with various time-lag relationships [127]. One of the most widely uses of time series analysis is to forecast future values.

For example, time series analysis was used to evaluate human mobility [128] using the number of available bikes in stations in Barcelona. After detecting temporal and geographical mobility patterns within the city, time series analysis was used to predict the number of available bikes for any station. Another study [129] showed how monthly time-series data could be used to explain aggregate demand for public transit in particular areas based on the prices of private and public transport, service characteristics, comfort levels, etc. The study was completed for Montreal and the results showed that the model could be used to predict the monthly number of adults likely to use the Montreal Urban Community Transit Commission (MUCTC) services. In another study, the temporal aspects of the relationship between petrol prices and public transport in US cities were analysed for the period 2002–2009 [130]. Time-series analysis was used to estimate any presence of lagged effects of price and service on transit patronage.

### 4.6.3. Clustering and Classification Methods

Data clustering is an unsupervised classification of patterns in a dataset. In transport applications, the clustering could include classifying the data into groups and creating general representations that are useful to optimise public transport services [131,132]. For example, access to smart card data from automated fare collection systems has been used to analyse passenger behaviour and resulted in classifications of groups of passengers that depict similar behaviours [133]. Clustering methods fall into several categories such as space partitioning, hierarchical methods, and density-based methods. One of the most commonly used space partitioning methods is the k-means algorithm, which is used to cluster n number of objects into k partitions where k < n. In a study to predict online bus arrival time using k-means cluster algorithm [134], the authors introduced a method by developing delay data clusters in accordance with the delay and time of the day. Next, hierarchical agglomerative clustering (HAC) was used to develop a hierarchy tree, by progressively merging clusters from the individual elements. The HAC is used to study passenger weekday profiles in order to uncover patterns in passenger travel behaviour [135,136]. The DBSCAN is a commonly used density-based algorithm, which is capable of identifying the common groups in a large spatial dataset through observing the local density of corresponding elements [137]. Moreover, k-means and DBSCAN clustering methods have been used to identify passenger boarding and alighting times, and location by analysing the smart card data [138–142].

Classification techniques are useful to discern patterns in the data as well as predict a future outcome based on historical observations [143]. Some of the other well-known classification techniques include the naive Bayes classifier, decision trees (DT), artificial neural networks (ANN), and support vector machines (SVM) [144]. Probabilistic and k-nearest neighbour classification models have also been used to classify activity-based travel choice patterns, which are useful to cluster users based on their activity patterns [145]. AI tools can be used to extract important features and predict quantitative and qualitative transport data in real time [146]. Clustering and classification [147,148] analysis methods are used to uncover hidden patterns in streams of data. For example, clustering and classification methods have been used [149] to study train station crowd patterns with the use of smart card data. The authors used HAC and dynamic time warping (DTW) clustering methods to regroup stations based on passenger usage patterns and then introduced three classification techniques

to forecast the crowdedness of stations. These advanced data analysis methods have been shown to be useful for knowledge extraction when the data includes rich information on group behaviour.

### 4.6.4. Machine Learning

Machine learning can help to detect spatial and temporal features from transport data [150,151]. Spatial data are usually data identified within a location such as information on the public transport routing, stops, and road network. While temporal data represent a time state data that could include, weather information, traffic conditions, timetables etc. The algorithms used in MaaS, for example, include co-training, co-regularisation, and margin-consistency style algorithms. These are generally referred to as multi-view learning algorithms as noted by [152]. Also, machine learning can help to detect the relationship between cause and effect from variables and also to find a common behaviour that two variables share [153]. Machine learning has also been used to determine parameters that affect the development of transport networks around the city such as the behaviour of network users, peak hours, and incidents during the day. This variation can be recognised to ensure that the users of that network have safer and more convenient trips with fewer delays.

In another study [154], the authors tried to understand the behaviour of public transport users through their smartphones. The study used information from the Global Positioning System (GPS), Geographic Information System (GIS), and sensors on the smartphone and combined it with machine learning techniques to detect and classify people's mode of transport, i.e., walking, train, bus, and taxi. The authors used different machine learning techniques and achieved an accuracy of 95% and above [155,156]. AI and machine learning techniques have helped in launching different MaaS mobile applications, worldwide. For example, the Whim app in Finland obtains data on people's preferred mode of transport and suggests the best way to get to a destination by booking online through a different mode of transport. Moreover, Qixxit and Moovel in Germany, Beeline in Singapore, and Ubigo in Sweden share similar techniques [157–160].

### 4.6.5. Big Data Analytics

Big data analytics are used to analyse data that is characterised as being unstructured, vast, and fast moving which is difficult to manage using traditional methods. Big Data tools include key technologies like Hadoop, NoSQL, MongoDB, and HDFS. These tools are applied to extract valuable information from data [161,162]. These techniques have been applied successfully in marketing, fraud detection, risks quantification, automated decisions for real-time processes, better planning, and forecasting [163,164]

A recent study has explored smart cities applications of Big Data by proposing an analysis service based on cloud technology [165]. The authors suggest that these can be further developed to generate information intelligence and support decision-making in smart future cities [166]. The authors presented a business model of Big Data for smart cities, which they believe could be used as a benchmark for future development of smart cities.

### 4.6.6. Optimisation Techniques

Optimisation problems are often complex and solutions are not readily obtained using a direct approach [167,168]. A major contributing factor to the complexity of a problem at hand is the number of decision variables [167]. There are numbers of approaches to solving large-scale optimisation problems which have a large number of decision variables [169]. Some of the approaches include stochastic optimisation, robust optimisation, simulated annealing, and convex–concave regularisation [170–174]

Stochastic optimisation is a collection of methods to minimise or maximise an objective function when there is randomness. In the past few decades, these methods have been useful in applications related to many fields such as science, engineering, business, statistics, and computer science [175]. Four of the broadly used methods are as follows: (i) sample average approximation, which is a two-part method dealing with sampling and deterministic optimisation [176,177]; (ii) stochastic approximation,

which is an iterative method which uses noisy observations to find the root of a function [178]; (iii) response surfaces is a set of methods which fit a surface to a set of decision-response pairs and search the surface to derive a new decision [179,180]; and (iv) global meta-model optimisation, which relates to an expected output through a set of inputs through regression [181]. Stochastic optimisation is an area with much active research in transport. For example, [182] has used the stochastic quasi-Newton method to produce a destination choice model with pairwise district-level constants for trip distribution based on a nearly complete regional OD trip matrix.

Metaheuristic optimisation is related to optimisation problems using metaheuristic algorithms. These metaheuristic algorithms are generally nature-inspired. From simulated annealing to ant colony optimisation, and from particle swarm optimisation to cuckoo search, many new metaheuristic algorithms have developed in all areas of optimisation [183]. Metaheuristics are useful in optimising block-box systems, where no gradient information and explicit information are available [184].

A solution to an optimisation model is defined as 'solution robust' if it lays near optimal for all scenarios of input data, and 'model robust' if it lays almost optimal for all data scenarios. Such a model formulation is defined as robust optimisation [185]. Robust optimisation is generally used to plan large-scale systems that are subjected to noisy, incomplete or uncertain input data [186]. It is widely used in many real-world problem domains [185,187] Reference [188] proposed a novel and reliable bus route schedule design problem using robust optimisation. The authors accounted for the uncertainty of bus travel time and the bus drivers' schedule recovery efforts. They developed a robust optimisation model to minimise the product of a weighting value and the sum of the expected value of the random schedule deviation and its variability. They found that optimal scheduled travel time is dependent on bus drivers' schedule recovery behaviour, and decision makers' scheduling philosophies. Furthermore, [189] used a robust optimisation model to formulate and solve a bus transit network design problem. Through the robust optimisation, they aimed to minimise the product of a weighting value and the sum of the expected value of operator cost and its variability.

### 4.6.7. Deep Learning and Reinforcement Learning

Deep learning is a class of machine learning algorithms based on artificial neural networks. These are capable of learning the representations of data with multiple levels of abstraction. There are few types of deep learning architectures such as recurrent neural network (RNN) [190–193], convolutional neural network (CNN) [194–196], and deep belief net (DBN) [194]. Deep learning applications have grown rapidly in pattern recognition, signal processing, discrete choice modelling, and optimisation [197–201]. References [202–204] provide a good introduction to deep learning methods for learning of feature representation either supervised or unsupervised at more abstract and successively higher layers.

Transport-related problems such as, passenger flow prediction are complex and are usually non-linear problems that are influenced by many fixed and stochastic factors [205]. Recent advances such as powerful graphical processing units (GPUs) have enabled deep learning methods to exploit complicated, compositional, and non-linear functions effectively. These advances have also made it possible to learn hierarchical and distributed feature representations accurately and make effective use of labelled and unlabelled data [206]. Reference [206] developed a novel model for passenger flow prediction using deep learning methods. They used their model for Xiamen BRT stations to predict hourly passenger flow. The flow prediction model is combined with unsupervised training model based on stacked auto-encoder (SAE) and supervised training model based on deep neural network (DNN). Reference [207] also developed a deep-learning model to estimate bus passengers destinations with the use of entry-only smart card data and land-use characteristics.

Current research into solving public transport-related issues and developing mobility-on-demand services using deep learning methods are still at the early stage. The potential of reinforcement learning, which is a type of AI paradigm used to analyse interactions with the environment and learning from their mistakes, are very promising [208]. This can be used in route choice behaviour, which enables

the possibility of providing feedback on foregone or non-chosen alternatives [209]. Reinforcement learning is also an active research area in autonomous driving [210] and for predicting the demand for AMoD services.

## 5. Existing and Emerging Business Models

Vehicles in FMoD fleets can be categorised into a number of services as summarised in Table 4 and described below [19].

**Table 4.** Existing and emerging business models and characteristics.

| Model | Characteristics |
|---|---|
| Taxi services | • High cost<br>• Most conventional<br>• Highest personalized service |
| BRIDJ | • Uses real time traffic data and passenger inputs<br>• Ability to find fastest routes |
| UBER, LYFT, DiDi and Careem, Lift hero, Hop Skip Drive | • Shared service<br>• Available on demand<br>• Availability of digital platforms<br>• Less expensive<br>• Enhanced passenger safety and security |
| Van pooling (Uberpool and Lyftline) | • Available on demand<br>• Available through digital platforms<br>• Shared service<br>• Reduced travel cost |
| Optibus, Customized bus | • Personalized<br>• Flexible and demand responsive |

### 5.1. Shared Vehicles

Shared vehicle fleets have flexible schedules and routes, operate on-demand and focus on delivering the door-to-door convenience. Taxi services are the most conventional form of such business models and provide the highest personalised service for users, but comes at a high cost compared to other services. In more recent times, this business model had been challenged by the pioneers of shared on-demand services, including Uber and Lyft. The success of these companies has been facilitated by digital platforms that help users connect with drivers who offer different levels of service (e.g., Uber Black versus UberX). These companies have also recently introduced car-pooling services where passengers can share their rides with other passengers travelling in the same direction. The shared transport applications also include car sharing and bike sharing as described in the previous section.

As mentioned before, a key success factor of ridesharing is the plethora of web and smartphone-based digital platforms and solutions related to the transport sector [211]. Ride sourcing was initially defined as a type of ridesharing which provides services with the use of GIS and GPS technologies on internet-enabled mobile phones to order or organise real-time ridesharing [212]. Ride-matching software is used to match riders to drivers, automatically, with similar trips and notifications sent via smart devices [212]. Unlike conventional taxi services, passengers do not need to speak over a phone or hail a cab or even have the correct amount of cash ready for payment. They simply book their rides through a mobile application. The digital platform then matches drivers with travellers and handles payments. Although the taxi industries have now developed e-hailing services through mobile apps, they still operate on a conventional taxi company model and are more expensive than their competitors [213]. Uber (offering different services such as UberX, UberXL, and UberSelect) [214], Lyft in North America [215], DiDi in China and Australia [216], and Careem in the

Middle East [217] are among the most known ride sourcing service providers operating in many cities around the world. Although there have been some concerns about passenger and driver safety in these services, these companies are increasingly scrutinising their drivers and providing new functionalities in their apps to improve the safety and security of passengers and drivers. Other examples of these business models include Lift Hero in San Francisco [218] which was established to serve elderly and disabled passengers. In addition, in the same city and Los Angeles, there is a Hop Skip Drive [219] to facilitate children with their school rides.

Other applications include van ride splitting or pooling. These are interesting transport demand management tools which have the potential to reduce single-occupant vehicle trips [220]. It is now an emerging alternative mode of transport for commuters, particularly that they are available through digital platforms [221,222]. With this, the users get the opportunity to share the ride with other travellers whose trips are on similar routes and split the fare among each other; therefore, the travel cost per passenger is reduced. Services like UberPool and LyftLine have taken the lead to provide the service of ride splitting or pooling. Among the strategy that LyftLine uses is that they encourage the passengers to gather in a specific location in the city, and passengers are offered a discounted fare if they walk to the pick-up locations. Similarly, UberPool introduced smart routes with a similar strategy.

Demand-responsive shared transport (DRST), a service which is provided through a fleet of vehicles, booked via a mobile application by the users, and scheduled in real-time to pick up and drop passengers based on their needs. This service is a blend of individual door-to-door ride services and a typical transit system. Here, the operators' intention is to select an optimal approach to assigning vehicles to passenger's needs. From the users' perspective, there is a need to reduce additional travel time and distances that they have to experience due to shared rides, and possible reduced fees for individual rides. Recently, the efficiency of schedules of dynamic demand responsive transport (DRT) services are studied through simulation models. In this regard, agent-based models (ABM) were found to be effective in reproducing complex social systems and overcome certain limitations [223–225].

*5.2. Flexible On-Demand Mini-Buses*

Flexible on-demand mini-buses operate to deliver door-to-door convenience at a fraction of the cost of ordering a taxi. In recent years, there were many attempts and trials around the world to improve conventional bus services and provide convenient services to users through such business models. Examples include Optibus, which launched an AI-driven on-time optimisation solution [226]. China has also introduced a personalised and flexible demand responsive public transport service called customised bus (CB) [227,228].

One of the well-known on-demand public transport services is BRIDJ which was launched in 2014 in Boston, Massachusetts [229]. The service was subsequently expanded to Washington DC and Kansas City. This service uses real-time traffic data and passenger inputs to establish origin-destination data. The digital platform is capable of finding the fastest route and only stops at locations requested by a passenger to optimise the service. However, BRIDJ operation came to a halt in the US due to low ridership, the operation only during certain hours, lack of public awareness or marketing, and network-specific issues. Another application of on-demand minibus services is the Kutsuplus in Helsinki [230] which also came to end after a few years of operation, due to similar reasons.

It should be mentioned here that BRIDJ has been acquired by an Australian-based transit company and the system is currently being trialled in Sydney [231]. The key focus of the existing trial is to explore the feasibility of introducing a flexible, well-coordinated system operation as a reliable means of 'first and last kilometre' travel solution.

*5.3. Autonomous Mobility-on-Demand (AMoD)*

Autonomous mobility-on-demand systems are currently being promoted as a viable and cost-effective alternative to existing mobility-on-demand solutions [232]. Today, there are a number of trials around the world for delivering autonomous on-demand mobility. In the majority of cases,

a human backup driver is still required in the vehicle in case of emergencies. In their ultimate form, however, these solutions would not include humans or backup drivers and hence would result in a sharp reduction in travel costs for customers and would allow people to have more flexible mode choices [233–235].

Several recent studies, which relied on millennial surveys, report that younger people are less keen to own private cars. In a study by car sharing company Zipcar, it is reported that half of the millennials interviewed say they would prefer public transport and car sharing systems to privately owned cars [236]. With this in mind, shareable autonomous electric vehicles (particularly those in which electricity is produced through clean resources, e.g., wind turbines or solar systems) appear like a promising proposition for decreasing the overall number of private cars. This would in turn directly address the problems of oil dependency, pollution, promote higher utilisation rates and reduce parking lot sprawls [237]. To date, few studies have dealt with the implications of AMoD systems. Some of the studies of particular relevance to this research are described below.

### 5.3.1. Lisbon

The Lisbon study [238] examined the potential impacts that would result from the implementation of a shared and fully autonomous vehicle fleet. To perform this assessment, the researchers developed an agent-based model to simulate the behaviour of all entities in the system. Travellers, as potential users of the shared mobility system; cars, which are dynamically routed on the road network to pick-up and drop-off clients, or to move to, from, and between stations; and dispatcher system tasked with efficiently assigning cars to clients while respecting the defined service quality standards, e.g., with regard to waiting time and detour time. The analysis was based on a real urban context, the city of Lisbon, Portugal. The simulation used a representation of the street network, using origin and destination data derived from a fine-grained database of trips based on a detailed travel survey. Trips were allocated to different modes: walking, shared self-driving vehicles, or high-capacity public transport. A set of constraints were established (e.g., that all trips should take at most five minutes longer than today's car trips take for all scenarios, and assumed all trips are done by shared vehicles and none by buses or private cars). The study also modelled a scenario, which included high-capacity public transport (Metro in the case of Lisbon). The study modelled two different car-sharing concepts, "TaxiBots", a term the researchers coined for self-driving vehicles shared simultaneously by several passengers (i.e., ride sharing), and "AutoVots", cars which pick-up and drop-off single passengers sequentially (car sharing). For the different scenarios, the researchers measured the number of cars, kilometres travelled, impacts on congestion, and impacts on parking space. The results indicated that shared self-driving fleets could deliver the same mobility as today with significantly fewer cars. When serviced by ride-sharing TaxiBots and a good underground system, 90% of cars could be removed from the city. Even in the scenario that least reduces the number of cars (AutoVots without underground), nearly half of all cars could be removed without impacting the level of service. Even at peak hours, only about one-third (35%) of today's cars would be needed on the roads (TaxiBots with underground), without reducing overall mobility. On-street parking could be totally removed with a fleet of shared self-driving cars, allowing in a medium-sized European city such as Lisbon, reallocating 1.5 million square metres to other public uses. This equates to almost 20% of the surface of the kerb-to-kerb street area (or 210 football pitches!). These findings suggest that shared self-driving fleets could significantly reduce congestion. In terms of environmental impact, only 2% more vehicles would be needed for a fleet of cleaner, electric, shared self-driving vehicles, to compensate for reduced range and battery charging time.

### 5.3.2. Stockholm

In the Stockholm study [239], the assessments included both a fleet consisting of currently in use gasoline and diesel cars as well as electric cars. The results showed that an autonomous vehicle-based personal transport system has the potential to provide an on-demand door-to-door transport with a

high level of service, using less than 10% of today's private cars and parking places. In order to provide an environmental benefit and lower congestion, the autonomous vehicle would require users to accept ride-sharing, allowing a maximum 30% increase of their travel time (15% on average) and a start time window of 10 min. In a scenario where users were not inclined to accept a lower level of service, i.e., no ride-sharing and no delay, empty vehicle drive will lead to increased road traffic increasing environmental impacts and congestion. In a scenario which looked at electric cars, an autonomous vehicle-based system and electric vehicle technology seemed to provide a 'perfect' match that could contribute to a sustainable transport system in Stockholm.

### 5.3.3. Austin

The Austin case study [240] investigated the potential travel and environmental implications of autonomous shared mobility systems by simulating a 12-mile by 24-mile area in Austin, Texas. The multi-agent transport simulation (Matsim) software was used for conducting this experiment using 100,000 randomly drawn person-trips out of 4.5 million Austin's regional trips. The study claimed that each autonomous shared car would almost replace around nine conventional vehicles within the 24-mile by 12-mile area while providing the same level of service, but would generate approximately 8% more vehicle-miles travelled. Their study also confirmed that this system would decrease the emissions by not only replacing the heavier vehicles with higher emissions rates but also by cutting down on the number of cold starts.

### 5.3.4. New York

The New York case study [241] introduced the expand and target algorithm, which was integrated with three different scheduling strategies for dispatching autonomous vehicles. The study also implemented an agent-based simulation platform and empirically evaluated the proposed approaches using New York City taxi data. Experimental results demonstrated that the algorithms significantly improve passengers' experience by reducing the average passenger waiting time by around 30% and increasing the trip success rate by around 8%.

### 5.3.5. Melbourne

In a similar study conducted in Australia [242,243], the authors explored the performance of Autonomous Mobility-on-Demand (AMoD) systems under uncertain travel demands in an urban environment using a case study of Melbourne. The results of the simulation model developed for the study showed that an AMoD system could reduce the current fleet size by 84% while still meeting the same demand for travel. This, however, comes at a cost of more vehicle-kilometres Travelled (VKT). The increase in VKT is significant and amounts to around 77% for scenarios in which the vehicles are used in car-sharing systems, and 29% for the scenarios in which vehicles are used as ride-sharing systems. These findings showed that the benefits reported in other studies have mainly been overestimated. In this study, the authors also discovered a strong quadratic relationship between AMoD fleet size and VKT [242].

## 6. Challenges and Opportunities

It is expected that operating public transport services in a flexible, on-demand manner would lead to higher user acceptance and optimum network utilisation. In order to achieve the above, there are a few critical obstacles to overcome. The main challenge of FMoD service optimisation is to find the trade-off between obtaining the profitability level for operators while maintaining high-quality service for users [19]. To accomplish this, future prediction of passenger demand to the FMoD services, adequate vehicle fleet allocation, and optimum route choice need to be conducted in real-time. Real-time big data analysis for traffic flow prediction is a key challenge [244], which is further complicated by unpredictable human behaviour making a future prediction of users' travel patterns even more challenging.

Other challenges include safety concerns, liability in case of an accident and bootstrapping problems, which is if there are many users for a particular ride-sharing service, there are many opportunities to find users that can share a ride [95]. Ridesharing is generally characterised by the following features: dynamic, independent, cost-sharing, non-recurring and prearranged trips. The main aims of implementing a system like a ride-sharing are the minimisation of system-wide vehicle miles and travel time, and maximisation of the number of participants [245]. Therefore, addressing these issues and improving the system to achieve the objectives are challenging. Furthermore, [245] has discussed three ride-sharing variants, namely single rider-single driver arrangement, single driver-multiple rider arrangement, and single rider-multiple driver arrangement. To obtain optimal arrangements for the above, deep learning theories and advanced mathematical models are needed but these are increasingly being developed with a higher level of detail and sophistication and once well-developed will overcome many of the existing challenges today.

Furthermore, the ride-sharing or ride-hailing services such as Uber and Lyft have also been criticised as a threat to public transport (particularly suburban buses) as these services are cannibalising other forms of conventional public transport modes. The bus industry, in particular, is increasingly being asked to innovate and improve its offerings to be able to compete with other more convenient modes.

The need for further research and in particular to develop a better understanding of the potential impacts of these services on congestion, the environment and the wider social impacts is needed. Further research on user acceptance is also required, particularly circumstances under which users would use such services, especially autonomous solutions in the future, and how they will impact land use changes.

Autonomous solutions introduce some other technical and operational challenges [16,246]:

1.  Minimum fleet sizing: This relates to determining the minimum number of vehicles required to keep outstanding demands uniformly bounded. Parameters such as arrival rate, average O-D distance, mobility demand distribution, average velocity and average service times are important to consider.
2.  Performance-driven fleet sizing: The number of vehicles to be used to ensure the quality of service provided to the user is no less than a given threshold.
3.  The ability of autonomous vehicles to gain environmental awareness to enable reliable, smooth, and safe driving.
4.  Challenges when connecting intelligent vehicles and infrastructure to provide shared, on-demand service to the customers.
5.  Software challenges, especially system security and integrity.
6.  Other social, user acceptance, planning, standards, legislative and insurance challenges with regard to the operation of AMoD

A thorough understanding of the above challenges of FMoD is important to embrace the future of mobility to welcome every potential opportunity.

*Addressing the Challenges of Flexible Mobility on Demand (FMoD)*

There is strong research momentum today to address the key challenges facing these novel mobility solutions [247]. In particular, there is an increasing focus on the role of vehicle automation in developing next-generation on-demand services. For example, a recent study on using electric vehicles [248] has proposed a fleet optimisation and operational model which was evaluated in simulation.

In another study, an innovative interface has been developed [249] for on-demand bus systems, where users can register or nominate bus stops themselves. In addition, [250,251] described an innovative on-demand bus system development and validity for different city types in Japan. Using machine learning techniques, Q-learning [252] has found an effective pick-up point selection process.

In the operation of the on-demand bus system, buses move only when a passenger requests the service through the internet or mobile phone. The request includes details of pick-up and drop-off location and the desired time of arriving at a destination. This is made possible through the selection of most appropriate nearby vehicles and routing algorithms. The algorithms would first determine which buses should accept the new service request. Then, the routing algorithm is used to identify the route and to schedule based on the new service requests [253].

Furthermore, operational level cost-effective services comprise determination of adequate fleet size followed by optimal operation of buses. Therefore, passenger demand prediction is important to determine bus headways and the required fleet size. Therefore, passenger demand prediction models developed by [254,255] serve as good examples but these need to be supplemented by a deeper understanding of the travel patterns [256].

The authors of this paper are currently engaged in research to develop and evaluate practical models to deliver convenient, efficient, reliable, and door-to-door on-demand transport service. The work is being supplemented by comprehensive simulation models that will be used as test-beds for evaluating the performance of these enhanced algorithms.

## 7. Policy Insights

The MaaS and shared mobility trends are breaking down the boundaries between different transport modes. This is largely due to the fact that technology is creating an intermediate level between the different means of transport and their users, which is made possible by the fusion of data into a new data layer. This is introducing new challenges to the regulatory and policy frameworks of transport. For the users, the focus will no longer be on the transport mode, but rather on mobility which will increasingly become an information service with physical transportation products, rather than a transportation product with services [257]. In addition to ensuring public safety and access to services, regulators and policymakers must also oversee other aspects such as interconnection, interoperability, capacity management, standards, and security. To achieve this, regulators must focus on the establishment of new comprehensive regulatory frameworks, which would enable the use of information technology, especially across the different transport modes. In particular, this will imply the need to develop an outcome-focused regulatory framework, which (1) puts the users at the centre of the new mobility system; and (2) encourages innovation and promotes safety, security, social equity and environmental sustainability. In the short-term, MaaS will also provide public agencies with the opportunity to bring innovation to their own transport services. In the long term, public agencies may need to rethink their role and consider opportunities for public–private-partnerships and service agreements with private mobility providers (Center for Automated Research, 2016). This new perspective will allow policy maker to focus on the regulation of the new data layers as well as the interface between the data layers and the physical transportation services.

In many cities in the U.S., public agencies are increasingly considering MaaS as an opportunity to provide more mobility options and, also address the first and last kilometre problem, particularly during late hours of the night and in low-density areas. Few cities are already collaborating with mobility providers such as Bridj and Uber to supplement and strengthen their existing public transport offerings and are viewing the technology and mobility providers as partners rather than competitors. Some examples from Tampa, Dallas, Atlanta, Memphis, and San Francisco demonstrate how this can be achievable and cost-effective [71].

Another challenge to policymakers is the protection of privacy. Those who have access to the data (and especially to the way the data is conveyed to the end-users) control the information and have immense power. The abuse of such data and information can result in market distortions, security risks, and diminished privacy protection [257].

The anticipated arrival of shared autonomous mobility-on-demand systems is also likely to introduce major challenges to policymakers. A two-day international workshop was conducted at the

Automated Vehicle Symposium in San Francisco, California in 2016. The aim was to develop a shared automated mobility service by utilising different studies and discussions worldwide.

Table 5 represents a summary of the implications of this technology regarding policies and future research needs [258]. Regulations will in the future play a key role in the emergence and development of autonomous vehicles and the new business models around them such autonomous MaaS. They are also likely to be the biggest hurdle for the deployment and public acceptance of these services. The regulators are already adapting and rethinking their approaches on this issue to avoid stifling the innovative uses of these technologies. Effective responses require an early and on-going dialogue between regulators, developers and the public in which regulators would create legal frameworks that are flexible but robust. An important role for the regulators will be to limit physical risks especially those that might be posed during interim years when legacy fleets of cars would interact with the autonomous vehicles that are offering the MaaS mobility solutions.

**Table 5.** A summary of the implications of the technology regarding policies and future research needs of shared automated mobility services [258].

| Implications in Terms of | Goals | Policy Changes/Solutions | Future Research Needs |
|---|---|---|---|
| **Effective automated shared mobility** | - Reduce costs, fewer delays, and improves users' experience | - Automated vehicle lane considerations<br>- Funding to adopt these new technologies<br>- Implement a single form of payment | - Payments on all types of shared mobility<br>- Implications related to labour and equity<br>- Design vehicles to reach the optimal level of services |
| **Safety measures** | - Reduce vehicle accidents/collision with other vehicles/ injuries and fatal. Improve safety in terms of preventing crimes<br>- Improve safety experience in the vehicles (e.g., harassment, antisocial behaviour, child safety) | - Implement safety targets and measures.<br>- Criteria for designing a vehicle (e.g., clear visibility, emergency button, surveillance) | - How to ensure safety on pickups points and drop-offs points<br>- Technology implementation that aid in avoiding collisions<br>- What are the acceptable collison rates?<br>- Different cultures present different safety standards |
| **Equity** | - Affordable services to all income levels<br>- Accessibility near schools, jobs<br>- 'Special needs' requirements<br>- Provide a free flow of data | - Require fare integration with equitable fare structures<br>- Equal allocations to transport networks to avoid congestion<br>- Effective road pricing<br>- Multiple pilot studies and testing<br>- Ensure a friendly infrastructure for automated vehicles | - Automated vehicles will lead labour issues?<br>- The transition between recent non-automated vehicles and automated vehicles.<br>- How to optimise the service/ search for optimisation methods. |

## 8. Summary, Findings, and Future Research Directions

Cities around the world are anticipated to benefit from the use of smart mobility technologies, but they must first overcome a number of challenges. The deployment of these technologies, complemented by appropriate governance and regulatory changes, will deliver substantial benefits [30]. These include user benefits such as personalised smart mobility services tailored to the user's diverse needs and easy access to mobility.

For the public sector, full deployment of technology-related infrastructure would improve the effectiveness of the whole transport system resulting in a more efficient allocation of resources,

improved traffic incident management and a more reliable transport system through advanced data processing.

For businesses, the benefits would include new profitable markets and business models for new transport services; renewed opportunities for the traditional transport and infrastructure business sectors as parts of innovative service concepts and co-operation; and smarter transport connections for all sectors (examples are Uber, Lyft, etc.).

Finally, it is worth noting that the sweeping changes anticipated by disruptive mobility have started to inspire visions of a very different future, as well as a good deal of hype. To distinguish between the hype and reality, it is important going forward to keep abreast of progress and advancements in this space, and to develop informed insights on how these disruptions are likely to change the urban mobility landscape and promote sustainable smart cities.

*Findings*

This paper provided a comprehensive review of the literature related to flexible mobility on demand systems, and included an environmental scan of the social, environmental, technological and economic impacts. The paper focused on the challenges and opportunities, and the impacts these systems are having on our cities, and success factors for their deployment. The paper first provided a historical overview of the developments in this field, describing applications that ranged from conventional dial-a-ride service to latest applications that rely on digital platforms and mobile apps. The paper also synthesised results from a large body of literature showing the advantages and other impacts of these mobility solutions, and the factors that led to either their failure or success. The paper described the numerous attempts and trials that have focused on changing travel behaviour and shifting travellers from car-dependent environments and encouraging the use of sustainable modes of transport. These studies documented some evidence of instances where users have sold a car or delayed purchasing a new one as a result of wide-spread shared mobility. Some of these findings are summarised in Figure 2 below.

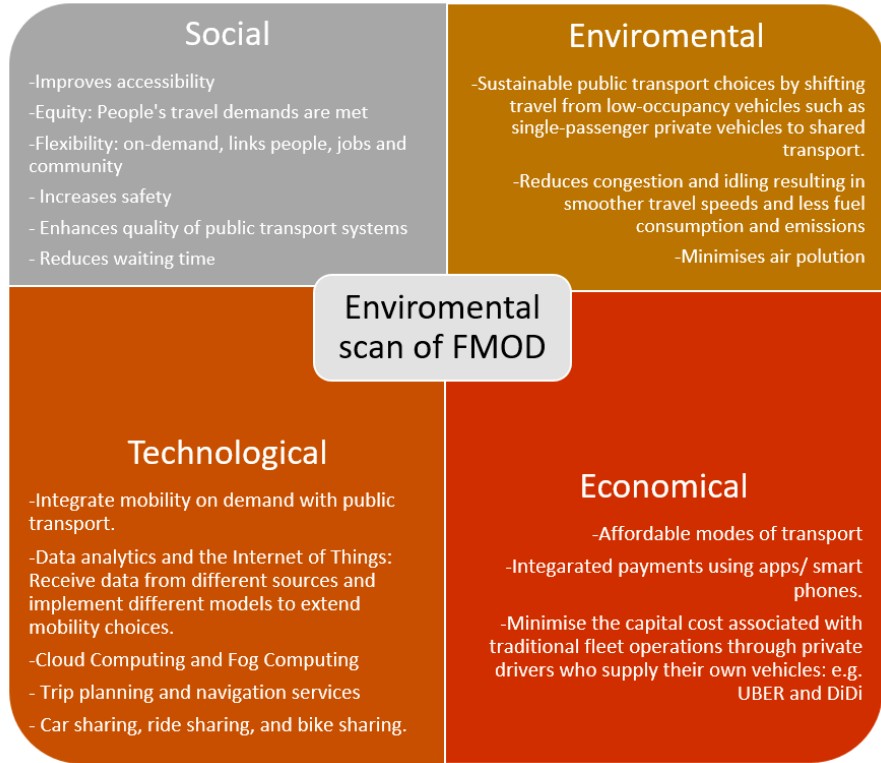

**Figure 2.** Environmental scan of FMoD systems. Source: Authors.

Despite these findings, more work is still needed, and this paper has identified the research challenges required to enhance the performance of these solutions from the commercial, operational, and public acceptance perspectives. In particular, the paper identified key factors to the success of these systems including the ability of underlying algorithms to predict travel demand patterns under uncertain demand profiles, which are made more complex given unpredictable human behaviour. There is also scope to improve the algorithms used for determination of optimum routes, and fleet and schedule management. To this end, machine learning, Big Data analytics, and large scale optimisation [259] are expected to dominate the field. Finally, and given these challenges, it is proposed that future research is informed by rigorous testing of these systems in simulation environments to test feasibility before commercial deployment in the field.

**Author Contributions:** The authors' contributions are as follows: Conceptualisation, S.L., H.D., and S.A.B.; Methodology, S.L., H.D., and S.A.B.; Writing—original draft preparation, S.L. and R.A.; Writing—review and editing, H.D. and S.A.B.; Supervision, H.D. and S.A.B.

**Funding:** This research received no external funding.

**Acknowledgments:** Sohani Liyanage acknowledges her PhD scholarship provided by the Swinburne University of Technology. Rusul Abduljabbar acknowledges the Iraqi Government for her PhD scholarship.

**Conflicts of Interest:** The authors declare no conflict of interest.

## Abbreviations:

| Abbreviation | Description |
| --- | --- |
| IoT | Internet of Things |
| FMoD | Flexible mobility on demand |
| DRT | Demand-responsive transport |
| DART | Dial a Ride Transit |
| FTS | Flexible transport services |
| MaaS | Mobility as a service |
| MoD | Mobility on demand |
| AMoD | Autonomous mobility on demand |
| AI | Artificial intelligence |
| MAV | Minimum attribute vehicle |
| STAR | Short term auto rental |
| B2C | Business to customer |
| P2P | Peer-to-peer |
| P2C | Peer-to-customer |

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
