# Peer review of "Flexible Mobility On-Demand: An Environmental Scan"

_sustainability, doi:10.3390/su11051262_

Reviewer 1 Report

I would like to thank the author for this article, I really found it very interesting and fully in line with the changes and advances in transportartion we are living in our daily lives in these days. Application of new technologies in transport on-demand services made it possible for this kind of transport to achieve a great level of flexibility, as you showed in the examples in your paper. 

I have just some small comments to suggest you:

Give another read to the whole paper, as there may be some little errors (e.g. in line 124 a stop point at the end of the sentence is missing)

I would add a resuming graph/image/table in the paragraph Existing and Emerging Business Models, poiting out their main characteristics

For yor classification of DRTS you can also have a look at  "Giuseppe Inturri, Nadia Giuffrida, Matteo Ignaccolo, Michela Le Pira, Alessandro Pluchino, Andrea Rapisarda. TESTING DEMAND RESPONSIVE SHARED TRANSPORT SERVICES VIA AGENT-BASED SIMULATIONS. In book: New Trends in Emerging Complex Real Life Problems"

Author Response

Manuscript ID: [sustainability-441155]:

Paper Title: [Flexible Mobility On-Demand: An Environmental Scan]

Authors: [Sohani Liyanage*, Hussein Dia*, Rusul Abduljabbar, Saeed Asadi Bagloee]

Dear Reviewer,

Firstly, we would like to thank you for the careful and thorough reading of our manuscript and for the thoughtful, extensive, meticulous and insightful comments and constructive suggestions, which help to improve the quality of the manuscript. Secondly, thank you for providing an opportunity to revise the paper.

We did our best to address all the comments. Below is a summary of the changes:

1.       Complete re-read was carried out and minor errors were rectified

2.       A summary table is added to the section of existing and emerging business models.

3.       Description of DRST was extracted from the recommended text and included in the paper under shared vehicles of existing and emerging business models.

We are providing the revised paper with every change highlighted in blue as well as point-to-point responses as following:

Point 1: Give another read to the whole paper, as there may be some little errors (e.g. in line 124 a stop point at the end of the sentence is missing

Response 1: Thank you. We have done a complete re-read to identify minor errors such as missing full-stops, commas, semi-colons, etc. These have been rectified in the revised submission. In addition, spelling mistakes and dashed words were corrected.

Point 2: I would add a resuming graph/ image/ table in the paragraph existing and emerging business models, pointing out their main characteristics

Response 2: Thank you. A table that summarises existing and emerging business models and their characteristics has been added to the section. The table is clearly highlighted in blue colour for your kind perusal.

Point 3: For your classification of DRST you can also have a look at “Giuseppe Inturri, Nadia Giuffrida, Matteo Ignaccolo, Michela Le Pira, Alessandro Pluchino, Andrea Rapisarda. TESTING DEMAND RESPONSIVE SHARED TRANSPORT SERVICES VIA AGENT-BASED SIMULATIONS. In book: New trends in emerging complex real-life problems.

Response 3: Thank you. The description of DRST was extracted from the recommended text and included in the paper under shared vehicles of existing and emerging business models. We hope this meets your requirement.

Yours faithfully,

Sohani Liyanage

Reviewer 2 Report

Thank you for adressing this very interesting problem. The paper provides really in-dept review of available research on this filed. However, quantity of information is also the major flaw of this contribution. As it is a review paper, it should follow some general rules which would make it informative. 

There is an inconsistency between the title and a structure of paper. An environmental scan is announced, but there is really any explanation how it is going to be proceeded.

Again, abstract informs about technological, social and economic impacts of shared mobility trends. They are scattered across the paper, but their identification is very difficult to a reader.

As this paper is a systematic/best evidence review, it should definitely contain a methods section. It should explain what evidence search strategy was selected, selection criteria etc. There should be an explanation how search procedure was connected to paper's objective(s). Without this, a paper tends to be just a list of referenced material. The method description would also be a help for an authors to better organise the contribution.

Lines #129-132 describing the paper's focus are loosely linked to the title and an abstract. This problem needs to be reconsidered, as well as paper's organisation.

A nature of a problem requires introduction of several terms and abbreviations: FMoD, MaS and so on. It is recommended to provide kind of an overview of these solution, as abundance of information makes it difficult to navigate between them. Maybe a well thought illustration?

Composition of case studies in section 2 is inconsistent with section 3 (Impacts....) Sec.2 focuses on car sharing and public transport innovations, while Sec. 3 tells a lot about bike sharing, MaasS etc. How impacts are assessed? This indicates lack of previously mentioned method of review description.

Section 4 is to my opinion the most interesting part of the contribution, provided it would be properly included into the paper. In fact only there there is a link to the title and environmental issues. 

It is recommended to revise an objective of the contribution. Having sec. 4 well prepared it could take a form of a revision on environmental influence of disruptive technologies in FMoD.

If above mentioned changes are followed followed, business models may be another pillar of review (objective), if supported with clear revision methodology.

Having suggested changes in mind, all sections following current Sec. 5 should be revised.

Author Response

Manuscript ID: [sustainability - 441155]:

Paper Title: [Flexible Mobility On-Demand: An Environmental Scan]

Authors: [Sohani Liyanage*, Hussein Dia*, Rusul Abduljabbar, Saeed Asadi Bagloee]

Dear Reviewer,

Firstly, we would like to thank you for the careful and thorough reading of our manuscript and for the thoughtful, extensive, meticulous and insightful comments and constructive suggestions, which help to improve the quality of the manuscript. Secondly, thank you for providing an opportunity to revise the paper.

We did our best to address all the comments. Below is a summary of the changes:

1.       Under the introduction section, we added details on how the environmental scan was carried out for this study and simultaneously a definition for an environmental scan was given. Here, a description was given for the search terms used and research databases used were included.

2.       A tabular illustration was included under the introduction section to introduce several terms and abbreviations such as FMoD, MaaS, etc.,

3.       A graphical illustration was included in the last section of to summarize the social, environmental, technological and economic aspects of FMoD.

4.     Titles of main sections and subsections were changed suitably for a better interpretation, understandability, consistency and to create a flow.

5.       Section 8.1 was added under the Summary, Findings and Future Research Directions section.

We are providing the revised paper with every change highlighted in blue as well as point-to-point responses as following:

Point 1: There is an inconsistency between the title and a structure of paper. An environmental scan is announced, but there is really any explanation how it is going to be proceeded

Response 1: Thank you. We have provided more details about the environmental scan, what it means and what is included in it. We also added some text to outline how we went about gathering the information for the literature review. Please see text highlighted in blue for further information.

Point 2: Again, abstract informs about technological, social and economic impacts of shared mobility trends. They are scattered across the paper, but their identification is very difficult to a reader.

Response 2: Thank you. We have now added a graphical illustration in the last section of the paper to summarize the key social, environmental, technological and economical aspects of FMoD. We hope this makes it easier for the reader to find the information more readily.

Point 3: As this paper is a systematic/best evidence review, it should definitely contain a methods section. It should explain what evidence search strategy was selected, selection criteria etc. There should be an explanation how search procedure was connected to paper's objective(s). Without this, a paper tends to be just a list of referenced material. The method description would also be a help for an author to better organize the contribution.

Response 3: Details of the methodology used for the study was added to the revised paper including the keywords used for the search, journals looked into and scientific databases searched.

Point 4: Lines #129-132 describing the paper's focus are loosely linked to the title and an abstract. This problem needs to be reconsidered, as well as paper's organization.  

Response 4: Thank you. The description of paper’s focus has been revised accordingly.

Point 5: A nature of a problem requires introduction of several terms and abbreviations: FMoD, MaaS and so on. It is recommended to provide kind of an overview of these solutions, as abundance of information makes it difficult to navigate between them. Maybe a well thought illustration?

Response 5: A table with all relevant abbreviations has been added to the revised paper. It introduces several terms and abbreviations such as FMoD, MaaS, etc.

Point 6: Composition of case studies in section 2 is inconsistent with section 3 (Impacts....) Sec.2 focuses on car sharing and public transport innovations, while Sec. 3 tells a lot about bike sharing, MasS etc. How impacts are assessed? This indicates lack of previously mentioned method of review description.

Response 6: Thank you. The titles were changed to suit what you suggested. This has allowed for a better interpretation, understanding, consistency and better flow.

Point 7: Section 4 is to my opinion the most interesting part of the contribution, provided it would be properly included into the paper. In fact, only there is a link to the title and environmental issues.

Response 7: Thank you. We trust that the changes made throughout the paper have helped with addressing this comment.

Point 8: Having suggested changes in mind, all sections following current Sec. 5 should be revised.

Response 8: Thank you. Section 8.1 was added and we trust that the changes throughout the paper meet your requirements.

Yours faithfully,

Sohani Liyanage

Round  2

Reviewer 2 Report

Dear Authors,

The revised version provides much better covergae of the subject. Considerable effort was made to addres such a broad issue.